# Circus Activities as a Health Intervention for Children, Youth, and Adolescents: A Scoping Review

**DOI:** 10.3390/jcm12052046

**Published:** 2023-03-04

**Authors:** Free Coulston, Kate L. Cameron, Kath Sellick, Madeline Cavallaro, Alicia Spittle, Rachel Toovey

**Affiliations:** 1Faculty of Medicine, Dentistry and Health Sciences, The University of Melbourne, Melbourne 3052, Australia; 2Clinical Sciences, Murdoch Children’s Research Institute, Melbourne 3052, Australia; 3Physiotherapy Department, St Vincent’s Hospital, Melbourne 3065, Australia; 4Newborn Services, The Royal Women’s Hospital, Melbourne 3052, Australia

**Keywords:** children, circus, paediatrics, scoping review, health outcomes, social-emotional health, physical health

## Abstract

Circus activities are emerging as an engaging and unique health intervention. This scoping review summarises the evidence on this topic for children and young people aged up to 24 years to map (a) participant characteristics, (b) intervention characteristics, (c) health and wellbeing outcomes, and (d) to identify evidence gaps. Using scoping review methodology, a systematic search of five databases and Google Scholar was conducted up to August 2022 for peer-reviewed and grey literature. Fifty-seven of 897 sources of evidence were included (42 unique interventions). Most interventions were undertaken with school-aged participants; however, four studies included participants with age ranges over 15 years. Interventions targeted both general populations and those with defined biopsychosocial challenges (e.g., cerebral palsy, mental illness, or homelessness). Most interventions utilised three or more circus disciplines and were undertaken in naturalistic leisure settings. Dosage could be calculated for 15 of the 42 interventions (range one-96 h). Improvements in physical and/or social-emotional outcomes were reported for all studies. There is emerging evidence of positive health outcomes resulting from circus activities used in general populations and those with defined biopsychosocial challenges. Future research should focus on detailed reporting of intervention elements and increasing the evidence base in preschool-aged children and within populations with the greatest need.

## 1. Introduction

The World Health Organisation’s (WHO) International Classification of Health Interventions defines a health intervention as “an act performed for, with, or on behalf of a person or a population whose purpose is to assess, improve, maintain, promote, or modify health, functioning, or health conditions” [1] (para.1). Knowledge and understanding of health interventions is essential to provide clinicians, policymakers, and service users with clarity regarding available options and the specific needs met by the interventions. Where new health interventions emerge, or activities transition to being used as a health intervention, it is essential to define the target (the body structure or function, activity, participation domain, or environmental factors being actioned by the intervention) [2], the outcomes and benefits, and the elements of the intervention itself [3]. This information may be used to inform intervention design, referrals, allocation of funding, and research [2,3]. Circus activities are one such emerging health intervention. Over the past few decades, circus activities have increasingly been used with children and young people to improve health and well-being [4], transitioning from a spectacle that people watch to an engaging recreational activity that people participate in [5]. As circus activities are increasingly being utilised to improve health and function in therapeutic [6,7], educational [8,9,10], and service settings [11], there is a clear need to define their use in this context.

The definition of participation in circus activities refers to the attendance and involvement [12] of young people in what have variously, and sometimes interchangeably, been described as social circus, community circus, youth circus, circus arts, and recreational circus [9,13,14,15]. Circus activities are typically described as having five distinct disciplines: aerials (e.g., trapeze), object manipulation (e.g., juggling), equilibristics (balancing activities such as tightwire), acrobatics (such as tumbling skills seen in gymnastics) and clowning [8,14,15]. The multi-disciplinary nature of circus presents an opportunity to focus many different activities towards a single goal or end-point, which may be particularly useful in a therapeutic context [16]. Furthermore, where children can participate in a variety of different physical activities, they may have greater opportunities for emotional regulation, identity exploration, and subsequent benefits to mental well-being [17]. Within each of the circus disciplines previously listed, there is the opportunity for a wide range of incremental challenges, meaning that participants can typically find an appropriate level of challenge, progress at their own pace and skill level, and experience early and frequent achievement leading to increased confidence and persistence [8,13,16,18,19,20,21,22].

As circus activities may present a rich and motivating health intervention opportunity, a comprehensive review of the elements of current circus interventions and their potential benefits is crucial and timely. Although a preparatory search of PROSPERO, MEDLINE, the Cochrane Database of Systematic Reviews and *JBI Evidence Synthesis* did not reveal current scoping or systematic reviews on this topic, it did reveal an emergent, heterogeneous body of literature unsuited to a more systematic review of the evidence. Therefore, the objective of this scoping review is to examine the extent and characteristics of the evidence on circus activities as a health intervention for children and young people aged up to 24 years, specifically (a) the characteristics of the participants targeted by the interventions, (b) the elements (frequency, intensity, time, type) and (c) health and wellbeing outcomes of the interventions, and (d) identification of gaps in the evidence to guide future research. This examination of the literature will allow researchers and practitioners to effectively advance the science of circus activities used as health interventions.

## 2. Materials and Methods

The review was conducted in accordance with the Johanna Briggs Institute (JBI) methodology for scoping reviews and the Preferred Reporting Items for Systematic Reviews and Meta-Analyses—Extension for Scoping Reviews (PRISMA-ScR) [23,24,25]. A protocol was developed and published by the authors a-priori to pre-define the objectives, methods, and proposed plan of the scoping review, as well as to ensure that the approach was systematic and appropriate both for the topic and the methodology [26]. A summary of the methods is provided below, with any modifications to the protocol justified and detailed.

This review is guided by the research question, “What is known about circus activities as a health intervention for children, youth, and adolescents?” Sub-questions of this review are:What participant characteristics (skill, developmental stage, or biopsychosocial challenges) are being targeted by the circus interventions?What are the key elements of the circus interventions (activities being taught; frequency, length, total number of sessions, and settings utilised)?What health and well-being outcomes are reported in the literature investigating circus interventions?What are the gaps in the current literature and how do they, and the findings of this scoping review, inform the planning of further research?

### 2.1. Eligibility Criteria

The criteria for inclusion of studies in this review have been developed based on the population, concept, and context framework related to the research questions [23]. Studies were included if participants were aged 0–24 years, or otherwise defined as children, youth, or adolescents [27,28], and the literature described or evaluated circus activities from the aerial, acrobatic, equilibristic, and manipulation categories taught to participants according to the WHO’s definition of a health intervention [1]. This review considered quantitative, qualitative, and mixed methods studies in any context to allow exploration of the variety of settings in which circus interventions are being conducted. In addition, grey literature, scoping, and systematic reviews were eligible for inclusion. No limits were placed on the publishing date, as the objective was to fully map existing literature in the area. Due to improvements in freely accessible translation services, language restrictions described in the protocol were removed, resulting in two sources of evidence [29,30] being translated into English using Google Translate (Google, Mountain View, CA, USA).

Studies were excluded if participants aged over 24 years were included, if age was not defined, or where the circus activity was not the therapeutic part (e.g., used as a warm-up only). Finally, interventions drawing solely from the “clowning” discipline of circus activities were excluded, as the evidence primarily sits in a clown doctor or clown care setting and would require a separate review.

### 2.2. Search Strategy & Study Selection

The search strategy was developed by the authors in consultation with a research librarian. A pilot search of two databases was undertaken, and additional text and index terms from identified articles were added to the search strategy (see Appendix A). To test the effectiveness of the strategy, the authors examined omitted records to ensure that no relevant studies were excluded.

Sources of information used in the final search (August 2022) were MEDLINE (Ovid: biomedical sciences), CINAHL Complete (EBSCO: nursing and allied health), Scopus (Elsevier: multidisciplinary), and PsycINFO (Ovid: behavioural science and mental health), ProQuest Dissertations and Theses Global, and Google Scholar. The strategy was adapted for each information source as required and reference lists of included sources of evidence were searched for additional eligible studies.

All identified records were collated and uploaded into the review management software Covidence (Veritas Health Innovation, Melbourne, Australia). Titles and abstracts were screened, and potentially relevant papers were retrieved in full and assessed in detail against the eligibility criteria. Screening was completed independently by three reviewers—F.C. screening all sources of evidence (100%), K.C. and M.C. independently screening 60% and 40%, respectively. Pilot testing to evaluate the reliability of the screening process showed 88% agreement between reviewers (75% was considered acceptable) [23]. Disagreements between reviewers were resolved through discussion with the third reviewer. Reasons for exclusion were recorded and presented in a PRISMA-ScR flow diagram (Figure 1).

### 2.3. Data Extraction and Analysis

A data extraction tool was developed a-priori (Appendix B) [26]. No modifications to the tool have been made post-protocol publication; however, due to reviewer availability, a decision was made by the review team to amend the extraction process. Data were extracted from 22 studies by two independent reviewers (F.C. and M.C) to improve rigour; however, after discussion to refine the extraction process, F.C. proceeded to extract all remaining sources of evidence (*n* = 35).

Extracted data were grouped under thematic headings relating to the review’s first three sub-questions. Data were then synthesised into a narrative summary with accompanying graphic or tabulated results. For the final sub-question, highlighted patterns, differences, and gaps in the findings across sources of evidence are explored in the Discussion, to guide future intervention development and research. Studies in a scoping review are not required to meet a quality threshold to be included, because the aim is to map and synthesise the current evidence, regardless of quality [23]. Therefore, no formal assessment of the methodological quality of the included studies was performed; however, key methodological issues affecting the validity and interpretation of findings are considered in the Discussion [31].

## 3. Results

### 3.1. Characteristics of Included Sources of Evidence

A PRISMA-ScR flow diagram (Figure 1) outlines the results of the search and screening process, including reasons for the exclusion of full-text papers [25]. An initial search conducted on 19 July 2021 resulted in a total of 723 citations, and a final search conducted on 4 August 2022 resulted in 128 citations. After de-duplication, 721 unique sources of evidence were screened at the title and abstract level, and 604 citations were excluded. The remaining 117 full texts were screened, and 41 sources of evidence were eligible for inclusion. After reviewing reference lists of included papers, another 16 sources of evidence were eligible, for the total inclusion of 57 sources of evidence.

Within these 57 sources of evidence, a total of 42 unique interventions are described, as multiple papers described single interventions (citations are grouped by intervention in Appendix C). Most sources of evidence (46/57) were published in the past decade, with 33% (*n* = 19) published in the past five years (Figure 2).

The majority of evidence on this topic is peer-reviewed (44/57), including a book [32] and book chapter [16], academic theses [20,29,30,33,34,35,36,37,38,39,40], and journal articles [4,6,8,9,14,17,18,21,22,41,42,43,44,45,46,47,48,49,50,51,52,53,54,55,56,57,58,59,60,61,62]. No systematic or scoping reviews fulfilled the eligibility criteria. Grey literature (13/57) included project reports [63,64,65,66,67,68,69,70], conference proceedings [71,72,73,74], and a newspaper article [75].

Study designs were described in 44 of the included sources of evidence and included: randomised comparison trials (2 unique interventions) [41,42,53,54,55,56,57,60,61], case studies [30,34,35,37,51,59,62], qualitative studies [14,21,22,29,38,40,45,46,68,73,74], mixed-methods [4,47,63,64,65,67], pre- post- designs [6,66], a cross-sectional design [17], a between-subjects repeated measures design [50], and a prospective, clustered quasi-experimental design (one unique intervention) [8,36]. Sample sizes ranged between one and 300 participants, and further details can be found in Appendix D.

### 3.2. Characteristics of Participants Targeted by Circus Interventions

This scoping review included participants aged up to 24 years, or otherwise defined as children, youth, or adolescents to allow examination of the developmental stage (e.g., pre-school), biopsychosocial challenge (e.g., cerebral palsy, mental illness, or homelessness) and/or skill (e.g., motor or social), being targeted by the circus interventions.

The development stages of participants in included studies may be grouped into pre-school (*n* = 3) [58,68,69], school-aged (*n* = 43) [6,8,9,14,16,18,20,22,30,32,33,34,36,37,38,39,41,42,44,45,47,48,49,50,51,52,53,54,55,56,57,60,61,62,63,66,67,69,70,73,74,75], and older adolescents and youth (*n* = 5) [4,17,30,35,46,64]. Age ranges larger than 15 years were present in four papers, with participants ranging from pre-school to post-school age [21,65,71,73]; six sources of evidence did not specify the age of the participants but described the inclusion of “children” [40,72], “youth” [43,64], or both [29,59]. Further details and mean ages (where they could be calculated) are presented in Appendix D.

Twenty of the included sources of evidence included participants from general populations. The remaining papers (37/57) included participants with defined biopsychosocial challenges such as cerebral palsy [41,42,45,46,48,53,54,55,56,57,60,61], vision impairment [69], a range of intellectual and/or physical disabilities [29,46,62], moderate to severe learning difficulties [65], overweight [49], developmental difficulties without a specified medical condition or disability [22], neurodiverse children and youth (including autism) [6,40,58,64,69], youth in residential psychiatric care [4,30,44,66,73], children and youth living in or exposed to warzones [21,35,72], and those described as “living on the street” [59], “at risk” [29,37,63,66], “vulnerable” [51,69], or as having “special needs” [69].

### 3.3. Key Elements of the Circus Interventions

To describe the circus activities being utilized as a health intervention (i.e., the “concept” of the scoping review [23]), the activities conducted in each intervention (the “What” [3]), the setting of the interventions (the “Where” [3]), and the dosage (“When and how much” [3]) are presented in detail in Appendix C. Only 15 of the 42 unique interventions fully describe each of these elements. The specific circus activities (the “What” [3]) can be categorised according to the five circus disciplines [26]. As can be seen in Table 1, 26 of the 38 interventions that described the circus activities conducted, draw from three or more circus disciplines, and five interventions draw from all five of the disciplines.

Figure 3 describes how often the individual circus activities were utilised in the interventions. Juggling and trapeze were utilised most often (in *n* = 30 and *n* = 18 interventions respectively). When looking at each circus discipline, object manipulation activities (which include juggling) were utilised most often in interventions (76 instances), equilibristic (balancing) activities were utilised in 54 instances, both aerial activities in 42 instances, acrobatic activities in 41 instances, and clowning in six interventions.

Figure 4 describes the “Where” [3] or context [23] of the interventions. More than half (27/42) of the unique interventions utilised naturalistic leisure settings, such as circus-specific centres [6,9,14,16,17,20,22,29,32,34,35,38,40,41,42,43,45,46,47,48,49,52,53,54,55,56,57,58,60,64,68,69,70,71,72] or community centres [4,21,51,72]. Educational settings (primary and secondary schools) were utilised in 16 unique interventions [8,9,17,18,33,36,37,39,50,62,63,65,66,67,69,72,73,74,75], and five sources of evidence describe interventions delivered in seven medical settings [30,44,66,69,73]. Eight sources of evidence reported multiple settings for the circus interventions [9,17,40,58,66,69,72,73]; three sources of evidence (two unique interventions) did not report the setting [43,59,70].

Table 1 illustrates the dosage (“When and how much” [3]) of the circus interventions. Circus sessions were most commonly delivered once (*n* = 19) or twice (*n* = 10) per week. Two interventions ran in an intensive model with five circus sessions per week for one [72], and two weeks respectively [41,42,45,48,53,54,55,56]. The duration of the circus activities ranged from 45-min to five hours, with a mode of one hour. Total intervention dosage ranged from 1–96 h in 15 interventions but could not be calculated in the remaining interventions due to a lack of detailed reporting. Sixteen studies (17 sources of evidence) investigated the effects of ongoing programs rather than a defined, separate intervention [14,20,29,30,35,37,38,40,43,44,51,52,62,68,69,70,71].

### 3.4. Outcomes Relating to Health & Wellbeing

Overall, the timing of outcome measurement was variable. For 24 interventions (31 sources of evidence), outcomes were assessed immediately following the intervention [8,9,22,34,36,47,49,50,53,54,55,56,57,60,61,63,64,65,66,67,70] or during the intervention [21,45,48], with three studies conducting additional mid-point data collection [6,46,70]. Five studies (nine sources of evidence) conducted additional three-month follow-up assessments [4,39,53,54,55,56,57,60,61], and one study (two sources of evidence) conducted 52-week follow-up assessments [60,61]. For participants in ongoing programs (12 sources of evidence), data was collected at a single timepoint in most cases [14,17,20,35,37,51,52,68,69,71]; however, one qualitative study collected data over a 12-month period [29], and yet another conducted a follow-up with participants four and five years after the initial data collection [38]. One study utilising outcome measures did not report the timepoint of data collection [59], and eight sources of evidence provided observations of circus activities used therapeutically with no assessment timepoints described [30,33,40,44,58,62,72,73,75].

The outcomes aiming to enhance health and well-being varied depending on the characteristics of the participants (as described earlier in this paper) but can be categorised into physical and/or social-emotional outcomes (see Figure 5 and Appendix E for details relating to extracted outcomes). Physical outcomes were targeted for children with cerebral palsy [41,42,45,48,53,54,55,56], those classified as overweight [49], as well as general populations [52]. Social-emotional outcomes were targeted in children exposed to warzones [21,35,72], those considered “at risk” [37,66], or “living on the street” [59], young people in residential psychiatric care [30,73] or with a range of intellectual [62] and/or physical disabilities [46], and those in general populations [9,14,17,33,34,39,43,47,67,70,71,75].

Both physical *and* social-emotional outcomes were targeted for children with vision impairment or “special needs” [69], children described as “vulnerable” [51], “at risk” [29,63] neurodiverse or children with autism [6,40,58,64], those with vision-impairment [69], with developmental difficulties [22], learning difficulties [65], in residential psychiatric care [4], as well as in general populations [8,18,20,36,38,50,68,74].

### 3.5. Physical Outcomes

Physical outcomes of circus interventions reported included increased participation in physical activity [8,18,36,50,74] or decreased inactivity [4], improved motor competence [8,18,36,69,74] or “physicality” [6], increased postural control [52], improved bimanual and occupational performance and unimanual upper-limb capacity [45,54,55,56,57], reduced lymphocyte proliferation, improved immune system function [49], as well as increases in fitness, strength, flexibility, coordination, body awareness [20,22,29,38,40,51,58,64,69], and acquisition of new physical skills [29,63,64,65,68].

### 3.6. Social-Emotional Outcomes

Social-emotional outcomes reported included building resilience and perseverance [14,21,36,39,48,60,64,70], developing confidence and/or self-esteem [8,9,14,18,20,21,22,30,35,36,40,46,47,51,62,63,64,65,66,67,68,69,74], enhancing development of social skills and connection [6,9,14,21,21,22,29,38,40,46,47,58,59,64,65,66,67,68,69,73,75], and fostering a sense of inclusion and belonging [21,29,35,58,59,67,68] including increased community participation [46]. Circus activities also resulted in increased intrinsic motivation, positive affect and grit [17], and were used to successfully build relationships in communities [33]. Positive impacts on mental well-being were reported [14,47], as was the growth of social-emotional skill development such as emotion management, teamwork, initiative, empathy, responsibility, and problem-solving [30,43,70]. Circus interventions resulted in improved communication [29,34,46,62,63,65,72] and emotional regulation skills [6,29,30,34], enhanced happiness or positive affect [20,21,69,71,73], and a decrease in destructive or “difficult” behaviours [37,68,72].

One study reported a decrease in the “emotional symptoms” subscale of the Strengths and Difficulties Questionnaire but no other differences were detected [50]. There were also no differences in psychosocial functioning for children participating in a circus program compared with those who were not [39], and in a qualitative study with young people in psychiatric care, participants reported no difference in their social skills, relationships, or self-esteem; however, the participants’ physicians reported substantial differences in the same constructs [4]. Another qualitative study reported improvements in children’s social-emotional development in circus class but noted that these improvements did not appear to cross over to schooling or home life [29].

The use of circus activities also increased motivation to participate in therapy [44,45] and physical activity [9,14,21,69,73] and improved quality of life for participants [60]. A Social Return on Investment analysis found “for every dollar invested, AUD$7 of social return may be generated due to participation in a circus program” [47] (p. 163).

## 4. Discussion

Circus activities are emerging as an engaging and unique health intervention [8]. Calls to define the use of circus activities in therapeutic contexts have come from circus practitioners themselves and more generally, as part of a movement toward clear, accessible intervention descriptions to inform future intervention design, referrals, allocation of funding, and research [2,3,6,30]. The current scoping review has identified an emerging evidence base of peer-reviewed and grey literature regarding the use of circus activities as a health intervention for children and young people. The final aim of this scoping review is to explore the implications and limitations of the synthesised evidence to inform the planning of further research.

Circuses’ potential for progressive challenge facilitates individual goal setting that aligns well with contemporary motor learning approaches, as does the focus on repetitive task practice within circus activities [16,20,76]. This person-centred approach means that circus activities can be modified to suit the child rather than requiring the child to change to suit the class [13,40]. Another person-centred aspect of circus activities is the appreciation of creativity, inventiveness and out-of-the-box approaches to skill accomplishment [20,77,78]. Encouraging self-expression and individual performance of the skill (as opposed to the *right* way to complete the skill) allows participants to accomplish tasks in ways that work for them and their bodies [18,35,77]. Participation in creative activities can have a positive effect on behaviour, self-confidence, self-esteem, relationship building, sense of belonging, and mental well-being [79,80,81]. Furthermore, the compatibility and collaboration of circus with other creative art forms, e.g., dance, theatre, music, fine arts adds to its appeal, as the hybrid nature of circus can provide “something for everyone” [14,15,18,19,20,35], [69] (p. 12).

Circus activities are also highly motivating due to their potential to satisfy the psychological needs of autonomy, competence, and relatedness [8]. Respecting participant preference and choice within the variety that circus activities offer may foster autonomy [16,17,42,48], and collaboration and cooperation with peers (as opposed to competition) to work toward shared goals in circus activities can support relatedness [16,17,48]. Lastly, circus activities can enhance participants’ sense of competence through the multitude of incremental challenges available within the multi-disciplinary offerings (e.g., aerials, object manipulation, equilibristics etc.). With appropriate scaffolding of activities, participants can find an appropriate level of challenge and experience early success [8,13,16,18,20,42,45]. When these psychological needs are satisfied, participants are more motivated and experience enhanced well-being, thus circus activities are an inherently satisfying and highly motivating intervention promoting participation, internal regulation, and persistence [48,82]. The unique community-based circus-specific centres that were the primary delivery sites in this review may also contribute to the motivating nature of circus activities, as research has demonstrated that children are more likely to participate longer and be more engaged when interventions are delivered in authentic, fun and novel settings [8,16,17,45,48,83]. All of these features align well within a biopsychosocial model of health, such as the WHO’s International Classification of Functioning, Disability and Health: children and youth version (ICF-CY) [16,84].

The ICF-CY framework draws connections between children’s participation, environment, personal factors, body structure, function, and activity, and their influence on health and development [84]. Rosenbaum and colleagues (2011) have further developed the ICF to provide a specific focus on six key areas (the F-words) of child development [85]. Interventions that can target multiple F-words have the potential to impact children’s health and development across multiple domains, and the findings of this review show promise for circus activities to enhance multiple physical and social-emotional outcomes in heterogeneous paediatric populations. Circus activities can be *fun* [45,46,68,77], they can increase physical and social-emotional *fitness* [20,22,29,30,40,69], and can be undertaken with peers (*friends*), providing opportunities for social connections [6,9,14,40,47]. Circus activities are often community-based and inclusive (*future*), and typically have a strong focus on person-centred *functioning*, that is, the individual performance of skills in a non-competitive environment [8,35]. They also have the potential to be *family*-centred, as a community-based recreational activity, and where a child and family’s goals can be incorporated into a fun recreational activity, positive therapy experiences have been reported [16].

Whilst circus interventions appear to have a role in all populations studied, increasing the evidence in populations with defined biopsychosocial challenges is recommended, as directing resources towards those most in need promotes equitable health outcomes [86]. Cerebral palsy was the most explored condition, with 15 sources of evidence (26%) describing circus activities conducted with this population. The majority of this evidence looked at young people aged 5–16 years with unilateratal hemiplegia who performed an intensive circus camp for 60 h, including 20 h of activities drawn from the aerial, acrobatics and object manipulation circus disciplines. Outcomes described in these sources of evidence align with the broader literature in this scoping review, with improvements described in both physical (bimanual and occupational performance, unimanual upper limb capacity) and social-emotional domains (engagement, motivation, confidence, social connection, and communication). For children with defined biopsychosocial challenges such as cerebral palsy, the provision of interventions early in life that show impact in physical and social-emotional domains aligns with literature on early intervention [87,88]. The potential impact of interventions is greatest in early childhood where rapid development and neural plasticity can be optimised to minimise neurodevelopmental impairments [87]. For example, long-term benefits such as improvements in academic outcomes, increased rates of employment and higher incomes have been demonstrated with early intervention in the preschool age group [88]. In the current review, only three studies looked at pre-school-aged children specifically, and only one of these looked at children with a defined biopsychosocial challenge. The scarcity of studies in early childhood, as well as the demonstrated benefits of early intervention may indicate a need for further studies in this population. Furthermore, although participant and intervention characteristics should vary depending on the aim of the study, individual and family-based goals are often related to the developmental stage and/or biopsychosocial challenges of the children [16]. Therefore, in future studies utilising participants across a >15-year range, or simply describing them as “children” or “youth” without an age specified, sampling should be strongly justified.

The positive results reported in all populations studied may indicate the potential generalisability of circus interventions; however, only seven studies involved long-term follow-up of outcomes. The inclusion of long-term follow-up is recommended in future research to determine retention of the positive results reported. Furthermore, future studies should determine the optimal intervention characteristics to improve these outcomes. With just over one third of the interventions providing a full description of the elements of the circus interventions, the use of the Template for Intervention Description and Replication (TIDieR) is recommended in future research to improve the reporting and replicability of circus interventions [3]. This encourages researchers to be clear about the essential elements of the interventions, thereby providing clarity for practitioners and service users. In those studies that did report the intervention content, most interventions drew from three or more circus disciplines, in most cases object manipulation, equilibristics, and aerial apparatus. This indicates that interventions contain activities that promote manual dexterity and hand-eye coordination, balance, and upper limb and core strength, respectively. There is evidence to suggest that the most common activity present in interventions, juggling, may improve neuroplasticity and reduce anxiety in adults [89,90]. Continuing to embrace the natural variety of circus activities in future research will ensure that “everyone is good at something” [69] (p. 18) and that participants can work toward individual goals while benefitting from social interaction with peers [8].

Regarding the limitations of the present review, scoping methodology does not seek to assess the quality of evidence and consequently cannot determine whether studies provide robust or generalisable findings or assess the “weight” of the evidence concerning particular studies [23]. Papers excluded in this review for having participants over the age of 24 years often also contained participants aged 18–24 years, which could account for the few studies included with participants in the 19–24 age group.

## 5. Conclusions

The findings of this review show promise for circus activities as a health intervention for children, youth and adolescents. This review found that the characteristics of participants described in the included sources of evidence were primarily school-aged, with fewer studies investigating participants from pre-school (0–5 years) and school-leaver (18–24 years) populations. Participants were drawn from both general populations and those with defined biopsychosocial challenges, and cerebral palsy was the most commonly described condition in the included literature.

The interventions described were most often delivered in authentic community-based settings and included activities from three or more of the circus disciplines; however, detailed reporting of intervention content was lacking. Outcomes of these interventions included improvements in both physical and social-emotional domains in general populations and those with defined biopsychosocial challenges; however, most studies did not include long-term follow-up.

Future research should focus on populations where the potential impact is greatest, such as pre-school-aged children with defined biopsychosocial challenges, and ensure long-term follow-up to understand retention of outcomes. Continuing to harness the unique motivating nature of circus activities, including variety, progressive challenge, and community-based settings is recommended. However, clear reporting of interventions using a framework such as the TIDieR checklist is essential to understand the crucial elements of circus interventions and investigate their impact on physical and social-emotional outcomes. These recommendations will enable practitioners and researchers to understand, replicate, and advance these approaches, and provide clarity for clinicians and service users regarding the use and impact of circus programs. Due to a lack of homogeneity in populations and study designs, a systematic review is not recommended at this time.

## Figures and Tables

**Figure 1 jcm-12-02046-f001:**
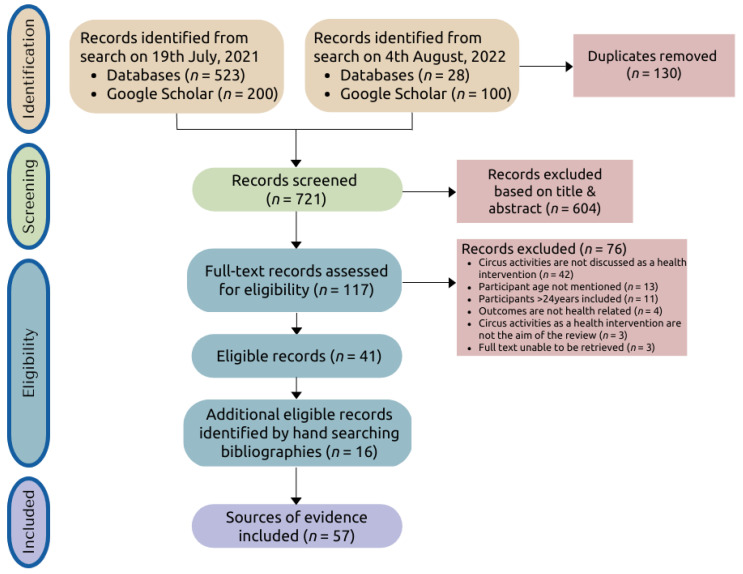
PRISMA flow diagram (adapted from [25]).

**Figure 2 jcm-12-02046-f002:**
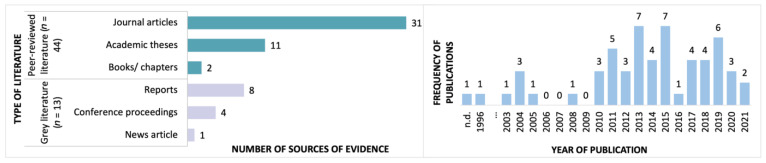
Year published and type of publication.

**Figure 3 jcm-12-02046-f003:**
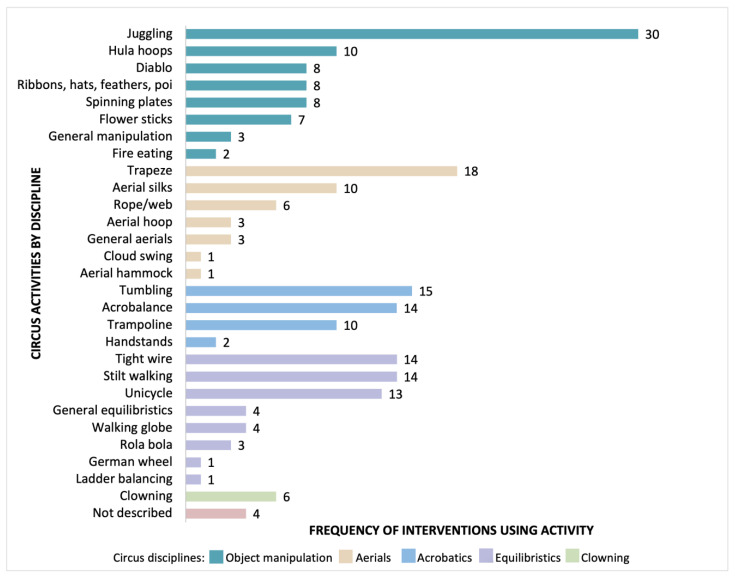
Circus activities utilised in interventions.

**Figure 4 jcm-12-02046-f004:**
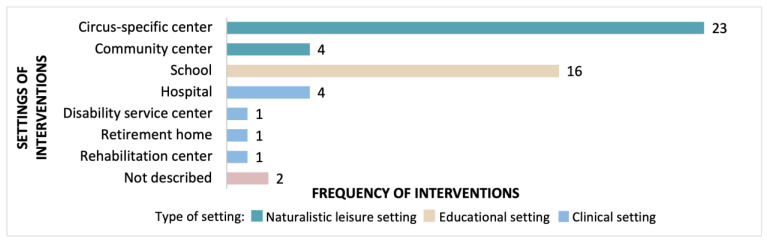
Settings utilised in circus interventions.

**Figure 5 jcm-12-02046-f005:**
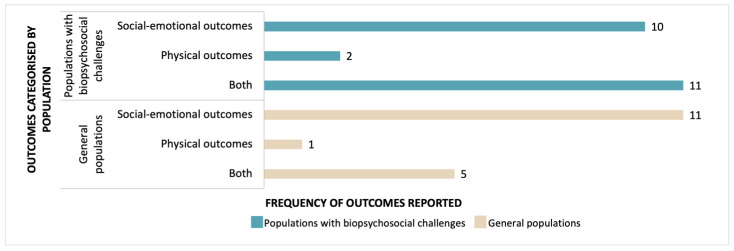
Categorisation of outcomes of circus interventions.

**Table 1 jcm-12-02046-t001:** Characteristics of the circus interventions.

	*Intervention Elements*				*Circus Disciplines*				
Citations Grouped by Intervention	*Sessions per Week* *(Number)*	*Total* *Sessions (Number)*	*Duration per* *Session (Hours)*	*Total* *Dosage (Hours)*	*Aerials*	*Acrobatics*	*Object* *Manipulation*	*Equilibristics*	*Clowning*
**Agans et al., 2019** [17]	Not described				X ^1^	X	X	X	X
**Barnett et al., 2020** [18]**; Kiez, 2015** [36]**; Kriellaars et al., 2019** [8]**; Valentini et al., 2020** [74]	1–4	10–84	1	10–84	X	X	X	X	
**Biquet, 2014** [35]	1	ongoing program	3		X	X	X	X	
**Bolton, 2004** [33]			2		-	X	X	X	
**Bonk, 2019** [34]	1	1	1	1	X	X	X	X	
**Boyd et al., 2010** [42]**; Gilmore et al., 2010** [45]**; Rodger & Kennedy-Behr, 2017** [32]**; Sakzewski et al., 2012** [60]**; Sakzewski, Provan, et al., 2015** [57]**; Sakzewski, Ziviani, Abbott, et al., 2011a** [54]**, 2011b** [55]**, 2011c** [61]**; Sakzewski, Ziviani, & Boyd, 2011** [53]**; Sakzewski, Ziviani & Poulsen, 2015** [16]**;**	5	10	2 (in addition to 4 h of other activities)	20 (of circus activities)	X	X	X		
**Boyd et al., 2013** [41]**; Miller et al., 2016** [48]**; Sakzewski, Miller, et al., 2015** [56]	5	10	2	20	X	X	X		
**Cadwell & Rooney, 2013** [71]		ongoing program			Specific circus activities not described				
**Caldwell, 1996** [75]	Not described						X		
**Candy, 2017** [20]	1 or 2	ongoing program	2.5		X		X	X	
**Cohen, 2018** [43]**; Smith et al., 2017** [70]		ongoing programs			Specific circus activities not described				
**Csuros, 2015** [44]	1	ongoing program				X	X	X	
**Fernandez et al., 2018** [22]	1	10	1	10	X	X			
**Fournier et al., 2014** [4]	2	24			Specific circus activities not described				
**Heller & Taglialatela, 2018** [6]	1	16	1	16	X		X	X	
**Kinnunen et al., 2013** [69]		ongoing program			X	X	X	X	
**Kovalenko, 2018** [29]	5	ongoing program	1.5–3			X	X		
**Loiselle et al., 2019** [46]	2	48	2	96	X	X	X	X	X
**Maglio & McKinstry, 2008** [9]	1	10–40	1	10–40		X	X	X	
**Mason, 2013** [72]	5	5				X	X		
**McCaffery, 2011** [65]	1	18	1.5–2	27–36			X	X	
**McCaffery, 2012** [64]		18	1.5	27			X	X	
**McCaffery, 2014** [63]	1	8	1.5	12			X	X	
**McCutcheon, 2003** [37]		ongoing programs			X	X	X	X	X
**McGrath & Stevens, 2019** [47]	1	20	1	20	X	X		X	
**Momesso dos Santos et al., 2015** [49]	2	38	1	38	X	X		X	
**Neave et al., 2020** [50]	1	18	1	18	X	X	X	X	
**O’Donnell, n.d.** [66]	1		1			X	X	X	X
**Ott, 2005** [38]	2	ongoing program	2		X	X	X	X	X
**Pompe, 2021** [30]		ongoing program				X	X		
**Rappaport, 2014** [73]	1–2		0.75–1.25		X	X	X	X	X
**Rivard et al., 2010** [51]	1–2	ongoing program				X	X		
**Sahli et al., 2013** [52]	2	ongoing program	2				X	X	
**Seay, 2004** [39]	Not described				X	X	X		
**Seymour, 2012** [40]	1	ongoing program			X	X	X	X	
**Seymour & Wise, 2017** [58]	1				X	X	X		
**Spiegel et al., 2015** [59]	Not described				Specific circus activities not described				
**Stevens et al., 2019** [14]		ongoing program			X	X		X	
**Taylor & Taylor, 2004** [62]	3	ongoing program	1				X	X	
**Trotman, 2013a** [67]	1	8	1.5	12		X	X	X	
**Trotman, 2013b** [68]	1	ongoing program	1				X	X	
**Van Es et al., 2021** [21]	2	16	5	80	X	X	X	X	

^1^ X denotes the circus activity was described as part of the intervention.

## Data Availability

The data presented in this study are available in the reference list.

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
