# Peer review of "Circus Activities as a Health Intervention for Children, Youth, and Adolescents: A Scoping Review"

_jcm, 2023, doi:10.3390/jcm12052046_

Round 1

Reviewer 1 Report

Dear Authors,

I really enjoyed reading this valuable article. I am very happy that my colleagues in Australia are interested in this issue. I think it is a very informative article on the psycho-social, physical and cognitive effects of circus activities. 

Best regards

Author Response

Dear reviewer, 

Thank you so much for your review of our manuscript, and your words of encouragement, 

Warm regards, 

The authors

Reviewer 2 Report

Review report

Comments to the authors

General comments

The article ‘Circus activities as a health intervention for children, youth, and adolescents: a scoping review’ is a well written paper that provides interesting and novel information that can contribute towards the current knowledge. However, there are some issues that the authors need to address.

Specific comments

Abstract:

Provide a clearly structured summary that includes (as applicable): background, objectives, eligibility criteria, sources of evidence, charting methods, results, and conclusions that relate to the review questions and objectives as PRISMA establishes.

Materials and Methods

Indicate in the manuscript the registration number of the protocol

Line 127: No limits were placed on the publishing date, as the objective was to fully map existing literature in the. This should be specified in selection criteria not in search strategy.

RESULTS

Flow chart should accompany the first paragraph of the results section (place after this paragraph as figure 1) to facilitate reader to follow the information given.

The figures are not clear and are difficult to read, please provide a better quality images/figures

In health interventions, one of the most important results of a review are the effectiveness of the interventions and the assessment tools. It would be recommended to provide a table that includes de variables assessed, the assessment tools used and the results achieved to summarise these aspects and facilitate the reader understanding the effects.

Discussion

I think that, since the special issue is ‘Novel Rehabilitation Approaches for Cerebral Palsy’ there should be more discussion related to this pathology in particular and the effects of circus interventions in these patients.

Conclusion

The conclusion should respond to all the objectives established (a) the characteristics of the participants targeted by the interventions (b) the elements (frequency, intensity, time, type) and (c) health and wellbeing effects of the interventions, and (d) identification of gaps in the evidence to guide future research. Please rewrite the conclusion so the conclusions respond to the questions.

As the conclusion talks about the effects of the intervention, ‘positive results were reported in both general populations and those with defined biopsychosocial challenges’, please specify these effects especially regarding cerebral palsy.

Author Response

Reviewer comments

Author responses

General comments

The article ‘Circus activities as a health intervention for children, youth, and adolescents: a scoping review’ is a well written paper that provides interesting and novel information that can contribute towards the current knowledge. However, there are some issues that the authors need to address.

Thank you for taking the time to comprehensively review our scoping review.

Abstract:

Provide a clearly structured summary that includes (as applicable): background, objectives, eligibility criteria, sources of evidence, charting methods, results, and conclusions that relate to the review questions and objectives as PRISMA establishes.

Thank you, as the journal requests, we have provided a single paragraph of 200 words maximum (without headings), including:

(1) Background: Place the question addressed in a broad context and highlight the purpose of the study;

Circus activities are emerging as an engaging and unique health intervention. This scoping review summarises the evidence on this topic for children and young people aged up to 24 years to map (a) participant characteristics (b) intervention characteristics (c) health and wellbeing outcomes, and (d) to identify evidence gaps.

(2) Methods: briefly describe the main methods or treatments applied;

Using scoping review methodology, a systematic search of five databases and Google Scholar was conducted up to August 2022 for peer-reviewed and grey literature.

(3) Results: summarize the article's main findings;

Fifty-seven of 897 sources of evidence were included (42 unique interventions). Most interventions were undertaken with school-aged participants, however, four studies included participants with age ranges over 15 years. Interventions targeted both general populations and those with defined biopsychosocial challenges. Most interventions utilised three or more circus disciplines and were undertaken in naturalistic leisure settings. Dosage was able to be calculated for 15 of the 40 interventions (range one-96 hours). Improvements in physical and/or social-emotional outcomes were reported for all studies.

(4) Conclusions: indicate the main conclusions or interpretations.

There is emerging evidence of positive health outcomes resulting from circus activities used in general populations and those with defined biopsychosocial challenges. Future research should focus on detailed reporting of intervention elements and increasing the evidence base in preschool-aged children and within populations with the greatest need.

Materials and Methods

Indicate in the manuscript the registration number of the protocol

Line 127: No limits were placed on the publishing date, as the objective was to fully map existing literature in the. This should be specified in selection criteria not in search strategy.

Thank you, the journal staff have indicated they accept published protocols (referenced on line 84) in lieu of registration.

As suggested, the content of line 127 has been moved to Section 2.1 "Eligibility Criteria".

RESULTS

Flow chart should accompany the first paragraph of the results section (place after this paragraph as figure 1) to facilitate reader to follow the information given.

Thank you, the PRISMA flow chart has been moved as suggested.

RESULTS

The figures are not clear and are difficult to read, please provide a better quality images/figures

Figure sizes have been adjusted for clarity.

RESULTS

In health interventions, one of the most important results of a review are the effectiveness of the interventions and the assessment tools. It would be recommended to provide a table that includes de variables assessed, the assessment tools used and the results achieved to summarise these aspects and facilitate the reader understanding the effects.

Thank you, a table has been included (Appendix E) outlining population, outcome measures, timepoints and results related to health & wellbeing outcomes. Although the question of effectiveness is extremely important, it is not one addressed by this scoping review, and would be best answered by a systematic review and meta-analysis. Due to the heterogeneity of the literature, we do not recommend a systematic review at this time.

Discussion

I think that, since the special issue is ‘Novel Rehabilitation Approaches for Cerebral Palsy’ there should be more discussion related to this pathology in particular and the effects of circus interventions in these patients.

Thank you, we have modified the discussion to incorporate a focus on the elements of interventions and outcomes reported for children with cerebral palsy.

Conclusion

The conclusion should respond to all the objectives established (a) the characteristics of the participants targeted by the interventions (b) the elements (frequency, intensity, time, type) and (c) health and wellbeing effects of the interventions, and (d) identification of gaps in the evidence to guide future research. Please rewrite the conclusion so the conclusions respond to the questions.

As the conclusion talks about the effects of the intervention, ‘positive results were reported in both general populations and those with defined biopsychosocial challenges’, please specify these effects especially regarding cerebral palsy.

Thank you, we have modified the conclusion to incorporate a focus on the elements of interventions and types of outcomes reported for children with cerebral palsy.

We have also modified the conclusion as suggested to draw on the main findings relating to the research questions while ensuring it remains concise.

Reviewer 3 Report

The paper submitted for review paper is a very interesting scoping review, while it is marginally relevant to the main topic of the special issue, which is "Novel Rehabilitation Approaches for Cerebral Palsy". The paper has an insignificant connection to an intervention directly addressing children with cerebral palsy. Also, authors should consider using Scopus and PEDro databases for their review. The use of PRISMA-ScR criteria is a strength of the work. Registering the review on the PROSPERO platform would increase the value of the work.

Author Response

Point 1:

The paper submitted for review paper is a very interesting scoping review, while it is marginally relevant to the main topic of the special issue, which is "Novel Rehabilitation Approaches for Cerebral Palsy". The paper has an insignificant connection to an intervention directly addressing children with cerebral palsy.

Response 1:

Thank you, we have modified the discussion & conclusion to incorporate a focus on the elements of interventions and outcomes reported for children with cerebral palsy.

Point 2: Also, authors should consider using Scopus and PEDro databases for their review.

Response 2: Although these are excellent databases, a search of these databases does not result in further articles that satisfy the inclusion criteria.

Point 3: The use of PRISMA-ScR criteria is a strength of the work. Registering the review on the PROSPERO platform would increase the value of the work.

Response 3: Unfortunately, PROSPERO does not accept scoping reviews for registration.

Round 2

Reviewer 3 Report

I am aware that this decision is disappointing for the authors but the paper is still marginally relevant to the main topic of the special issue, which is "Novel Rehabilitation Approaches for Cerebral Palsy". There is even no mention of CEREBRAL PALSY in the abstract so it should not be considered for this special issue from the very beginning - I mean that the assistant editors should not forward this paper for revisions. Unfortunately, this paper is without the scope of the main area of this special issue.

Author Response

Dear reviewer, 

Thank you so much for your response. As suggested by the Academic Editor, a line in the abstract now reads: "Interventions targeted both general populations and those with defined biopsychosocial challenges (e.g., cerebral palsy, mental illness, or homelessness)."

Many thanks for your time reviewing our paper, 

Warm regards, 

The Authors